SOFTWARE

# RNACOREX - RNA coregulatory network explorer and classifier

Aitor Oviedo-Madrid[1,2,3]*, José González-Gomariz[1,2,3], Ruben Armañanzas[1,2,3]

**1** Institute of Data Science and Artificial Intelligence, Universidad de Navarra, Pamplona, Spain,
**2** TECNUN School of Engineering, Universidad de Navarra, Donostia-San Sebastian, Spain, **3** Cancer Center, Clínica Universidad de Navarra, Pamplona, Spain

\* aoviedomadr@unav.es

## Abstract

Micro-RNAs (miRNA) and their relationship with messenger RNAs (mRNA) have been widely associated with disease development and progression. Post-transcriptional coregulatory networks are sets of miRNA-mRNA interactions that regulate specific genetic behaviors through their combined activity. However, identifying reliable sets of such interactions associated with specific diseases remains challenging, partly due to the high rate of false positives and the lack of user-friendly tools developed for this purpose. In this work, we introduce a new Python package called RNACOREX (RNA CORegulatory network EXplorer and classifier). RNACOREX is a new, easy-to-use tool that allows researchers to find disease associated post-transcriptional coregulatory networks and use them to classify new unseen observations of miRNA and mRNA quantifications. RNACOREX combines structural information from curated databases with expression data analysis, using conditional mutual information to infer reliable sets of miRNA–mRNA interactions. These sets are then used to build probabilistic models based on Conditional Linear Gaussian (CLG) classifiers, which allow both prediction on new samples and validation of the inferred networks.

To demonstrate its capabilities, we tested RNACOREX in 13 different databases from the The Cancer Genome Atlas Program, generating the associated post-transcriptional coregulatory networks and extracting classification performance metrics for each tumor type. Specifically, we used RNACOREX to classify patients according to their survival time in each cancer type, highlighting miRNA–mRNA interactions that consistently appeared across different cancer types. The results show that RNACOREX achieves competitive predictive performance compared to widely used classification algorithms, while offering the added benefit of interpretability through its graph-based modeling framework.

**Data availability statement:** Source code is available within the Github repository of the library:
https://github.com/ digital-medicine-research-group-UNAV/ RNACOREX (rnacorex 0.1.5). Specific data for replicating the paper's code from scratch is available at:
https://xenabrowser.net/datapages/?hub= https://gdc.xenahubs.net:443.

**Funding:** This work was partially supported by the Gobierno de Navarra through the ANDIA 2021 program (grant no. 0011-3947-2021-000023 to RA) and the ERA PerMed JTC2022 PORTRAIT project (grant no. 0011-2750-2022-000000 to RA). The funders had no role in study design, data collection and analysis, decision to publish, or preparation of the manuscript.

**Competing interests:** The authors have declared that no competing interests exist.

## Author summary

Cells regulate their behavior through complex molecular processes, many of which are controlled by small molecules called microRNAs (miRNAs). These miRNAs bind to messenger RNAs (mRNAs) and influence how genes are expressed. When this regulation is disrupted, it can lead to diseases such as cancer. Rather than acting alone, miRNAs and mRNAs often work together in groups, forming networks that jointly control gene activity. Understanding these networks can help us better detect, classify, and study diseases.

In this work, we introduce RNACOREX, a new computational tool that helps researchers discover these miRNA–mRNA networks associated with specific diseases. RNACOREX combines prior biological knowledge from trusted databases with real gene expression data to identify reliable interaction patterns. It then uses these patterns to build predictive models that can classify new patient data and highlight the most relevant molecular interactions for each disease.

We tested RNACOREX on 13 different cancer types using data from The Cancer Genome Atlas. The tool revealed disease-associated networks and achieved performance comparable to other popular machine learning models. While the current implementation does not yet provide definitive biological insights, it highlights molecular interactions most associated with each phenotype, pointing to a promising direction for future investigations connecting prediction with biological context.

## 1 Introduction

MicroRNAs (miRNAs) are small, non-coding RNA molecules that regulate gene expression by binding to target messenger RNAs (mRNAs), typically leading to their degradation or translational repression [1]. Through this mechanism, miRNAs influence a wide array of biological processes, including development, differentiation, and cancer progression. The set of miRNA-mRNA interactions form what are known as post-transcriptional networks, coregulatory layers that operate after gene transcription to modulate gene expression. Understanding these networks, particularly the specific interactions associated with certain diseases or biological conditions, can provide deep insight into the coregulatory patterns that underlie pathological processes.

The discovery and validation of a miRNA-mRNA interaction in a conventional laboratory requires large amounts of resources, making it unfeasible to study all possible interactions. To overcome this limitation, numerous miRNA target prediction computational methodologies have been developed, based on structural characteristics of the RNA sequences and statistical analyses of experimental data [2]. These methods have allowed the generation of databases of predicted interactions. However, a predicted interaction is not necessarily real, and even if it does exist, its biological relevance remains unknown. In this regard, there is a critical need for computational tools that not only predict reliable interactions, but also assess their functional relevance in the context of a given pathology or biological condition.

This challenge has traditionally been approached from two perspectives. Some methods leverage curated biological knowledge to define interaction sets, but these are generally not designed to use such sets as predictive models, limiting their practical validation outside the laboratory [3,4]. Conversely, Bayesian Networks (BNs) can be implemented to extract disease-associated sets of interactions combining both predictive power and explanatory capabilities [5]. General-purpose Bayesian network packages provide robust implementations of algorithms for network inference and prediction [6]. However, when these models incorporate no curated biological information, the resulting interaction sets may lose robustness and reliability, potentially leading to spurious or less biologically relevant connections (see S6 Text).

In this work, we present the RNACOREX package, a Python-based tool that allows researchers to unravel post-transcriptional coregulatory networks by identifying the most relevant miRNA-mRNA interactions associated with a specific disease or biological process. The novelty of RNACOREX lies in its integrated pipeline, specifically adapted to work with transcriptomic data, which enables researchers to easily identify relevant sets of interactions and use them to induce predictive models. The package uses a novel hybrid approach that integrates sequence and structure information from well-known databases with conditional mutual information based analyses of miRNA and mRNA expression data. This hybrid approach has demonstrated its ability to filter spurious interactions and identify more robust interaction sets [7,8]. The set of interactions is then modeled as Conditional Linear Gaussian (CLG) classifiers, a specific kind of BN classifier based on the graphical representation of the post-transcriptional coregulatory network. These models are particularly well-suited to this problem, as they avoid the need for data discretization and take advantage of the properties of Gaussian distributions to enable more efficient inference. This model is used to further validate the interaction set and classify new samples of miRNA-mRNA quantifications. In the current work, the RNACOREX pipeline was applied to 13 cancer datasets from The Cancer Genome Atlas, identifying cancer-associated interaction sets of interactions and using these sets to classify new samples.

## 2 Design and implementation

MicroRNAs regulate gene expression by binding to specific mRNAs, influencing their stability and translation. Analyzing the expression levels of both miRNAs and mRNAs linked to a particular phenotype can reveal key interactions that may contribute to disease progression. Structural features of miRNAs and mRNAs can also help identify candidate interactions. However, not all miRNAs interact with all mRNAs, and relying solely on expression or structural data often results in a high number of false positives [9]. RNACOREX addresses this challenge using a hybrid approach that combines functional information (based on empirical expression values) with structural information (including curated knowledge about the molecular structure of miRNAs and mRNAs). Functional and structural knowledge are translated into scores, which are then used to rank interactions according to their relation with the pathology.

### 2.1 RNACOREX pipeline

The pipeline of RNACOREX takes a dataset containing matched miRNA and mRNA expression profiles and a binary class or phenotype as input. Candidate interactions are initially filtered using curated prediction and validation databases, removing those that lack biological plausibility. For each remaining interaction, two complementary scores are computed: a *structural information score*, capturing the level of prior biological support for the interaction, and a *functional information score*, reflecting the empirical association between the interaction and the phenotype of interest, derived from expression data. These two scores are then integrated to establish an interaction hierarchy, which ranks miRNA–mRNA pairs according to both their biological credibility and functional relevance. Using this hierarchy, RNACOREX defines the topology of the Bayesian network and estimates the parameters of the CLG model. Once the model is built, the user can evaluate its performance and make predictions on new data. Fig 1 presents a graphical summary of the package workflow divided in its main seven stages.

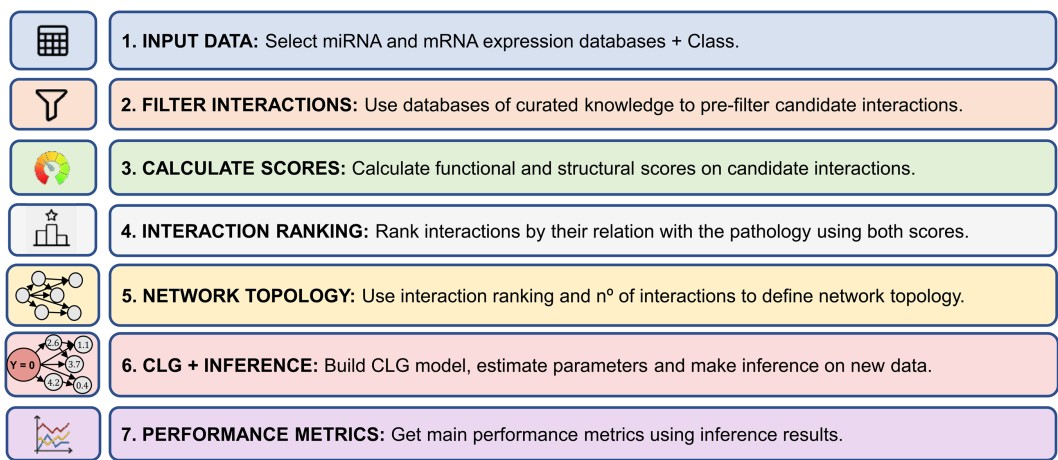

**Fig 1**. **RNACOREX package workflow.** (Own elaboration)

## 2.2 Interactions filtering

A growing number of computational methods have been developed to predict miRNA–target interactions (MTI). Among the most widely used resources for predicted MTIs are TargetScan v8.0 [10] and DIANA-microT 2023 [11], which base their predictions on sequence complementarity and other biologically relevant characteristics. In contrast, miRTarBase v10.0 [12] and TarBase v9.0 [13] compile MTIs that have been experimentally validated.

RNACOREX leverages these four databases to perform an initial biological plausibility filter. Specifically, it retains only those interactions that are supported by at least one of the previously mentioned sources (either as a prediction, or as experimental evidence) while discarding interactions absent from all of them. The resulting set of interactions defines the pool of candidate interactions **S** that RNACOREX considers for further modeling and analysis. The reasoning behind this is that if an interaction has not even been predicted using well-established features such as sequence complementarity, it is unlikely to occur in a real biological context. This filtering reduces the likelihood of including spurious interactions, though at the cost of overlooking unlikely novel ones. However, as many entries from TargetScan and DIANA remain computational predictions without experimental validation, recovering these interactions could guide future biological validation.

## 2.3 Information scores

For each interaction in the candidate set **S**, two metrics are computed: the *structural information score* and the *functional information score*. These metrics assess the relevance of each interaction based on its biological support and its association with the phenotype under study.

**2.3.1 Structural information score.** The structural information represents the biological support associated to each interaction. To quantify this relevance, the four databases used in the filtering step are considered: TargetScan, DIANA Micro-T, miRTarBase, and TarBase. Greater representation of an interaction across these databases will indicate stronger biological support, either because it has been consistently predicted, or because it has been directly validated through experimental evidence. The term 'structural' is used to denote that predictions generated by these tools are based on features such as sequence complementarity, regional accessibility, and target site position. If an interaction has been consistently predicted, it is likely due to its structural characteristics.

The structural information score *SI* is based on two components: the number of databases in which the interaction is present and the consistency index among the pairs of databases where it appears. The consistency index [14] measures the similarity between two sets of interactions, assigning greater weight to an interaction if it occurs in databases

that are themselves similar. The score is computed by summing the consistency indices of all pairs of databases in which the interaction is present. This index is normalized such that the total sum of indices across all database pairs equals 1, ensuring that the structural score of interactions present in all databases is equal to 1. The presence of a $(i,j)$ interaction in a $(S_k, S_l)$ pair of engines ($w_{ij}^{kl}$) will be represented as a binary value, being equal to 1 if the interaction is present in both engines and 0 if it doesn't.

$$SI_{ij} = \sum_{k=1}^{4} \sum_{l=1}^{4} I_{norm}(S_k, S_l) * w_{ij}^{kl}. \tag{1}$$

Interactions that appear only in a single engine may still remain at the stage of calculating structural information. However, since they are not supported by at least two engines, their structural information is automatically assigned to zero, as they cannot be considered as consistently predicted or validated interactions. Since the package is intended to serve as a tool for discovering new potential, not yet confirmed interactions, same weight is assigned to presence in validated and predicted databases. Nevertheless, validated interactions tend to be more consistently represented across all databases compared to predicted ones, thus benefiting from the consistency index and the structural score without the need for manual adjustment.

**2.3.2 Functional information score.** For all interactions with a non-zero *structural information score*, the *functional information score* is computed. This score, aims to include miRNA-mRNA relations empirically identified as relevant through their joint expression data. This association of the triplet miRNA-mRNA-class is analyzed in RNACOREX by computing the information that each miRNA-mRNA pair provides about the class being studied through their conditional mutual information (CMI) [15]. The value of the CMI for a $ij$ interaction regarding the class C is defined as the *functional information score* of the interaction ($FI_{ij}$).

$$FI_{ij} = CMI(i,j|C) = \sum_{c \in \Omega_c} \int_j \int_i p(c) f_{ij}(i,j|c) \log \frac{f_{ij}(i,j|c)}{f_i(i|c) f_j(j|c)} didj, \tag{2}$$

When calculating the conditional mutual information of each element, continuous expression values are kept, estimating their density functions with Gaussian kernel methods [16,17], and approximating the function's value using the trapezoidal rule [18]. Pairs with higher CMI values are expected to be more related with the class under study.

## 2.4 Interaction hierarchy

To balance existing structural and functional knowledge, the set of detected miRNA-mRNA interactions are separately ranked using their structural ($SI_{ij}$) and functional ($FI_{ij}$) information scores. These two rankings are then combined to determine the order in which interactions are included in the network. The top of the ranking includes those interactions with the greatest relevance with respect to the class under study and the highest biological significance. By default, RNACOREX applies the *"alternative"* ranking method to build the final hierarchy of interactions. In this approach, elements are selected alternately from the functional and structural rankings to generate the combined list. In cases where interactions have identical structural information scores, ties are resolved using the *"isolated"* method, giving priority to interactions composed of elements that occur less frequently within the pool of potential interactions, as these may represent a more isolated and potentially novel regulatory pathway. RNACOREX also supports other ranking strategies and tie-breaking methods, which are described in detail in S3 Text.

## 2.5 Conditional linear Gaussian classifier

A Bayesian network $B = (G, \theta)$ is defined by a Directed Acyclic Graph (DAG) $G = (V, A)$ where $V$ is the set of nodes and $A$ the set of directed edges, and a set of parameters $\theta$ representing the conditional probability distributions associated

with each node. A Conditional Linear Gaussian (CLG) model is a specific type of Bayesian network handling both continuous and discrete variables. In these models, continuous variables are modeled using Gaussian distributions whose parameters depend on the values of their parent nodes.

The networks inferred by RNACOREX consist of two types of variables: continuous expression variables (miRNAs and mRNAs) and a binary class variable (phenotype). This structure makes them ideally suited for representation using a Conditional Linear Gaussian (CLG) model, as it allows working directly with continuous data without the need for discretization, and enables highly efficient inference by leveraging the Gaussian properties of the variables.

In Conditional Linear Gaussian classifiers, the class variable $C$ is treated as a parent of all other variables; therefore, for a $X_i$ continuous variable, two types of dependencies are possible: (1) continuous variables that have only the class as their parent $pa_i = \{C\}$ (typically miRNAs), and (2) continuous variables that depend both on the class and on other continuous variables $\mathbf{pa}_i = \{\mathbf{Y}_i, C\}$ (typically mRNAs). If a variable depends only on the class, its distribution is modeled as a univariate Gaussian conditioned on the class: $P(X_i|pa_i) = P(X_i|C) \sim N(\mu(C), \sigma^2(C))$. If, in addition to the class, a variable has continuous parents, it is modeled as a conditional Gaussian distribution with a mean that depends linearly on the continuous parents: $P(X_i|\mathbf{pa_i}) = P(X|\mathbf{Y}_i, C) \sim N(\beta_0(C) + \beta^T(C) \cdot \mathbf{Y}_i, \sigma^2(C))$

The analysis using CLG classifiers can be divided into three main steps: (1) define the network topology, (2) estimate the network parameters, and (3) perform inference based on the network. The topology of the network is fully defined by the interaction ranking and the number of interactions ($k$) set by the user. The network will be composed of the first $k$ interactions in the ranking. Given a fully specified network structure, parameters are estimated from the expression data using the conditional (in)dependence relationships encoded in the topology. Finally, classification is performed by computing the joint likelihood of the observed data and selecting the class with the highest posterior probability:

$$c^* = argmax_{c \in \Omega_C} P(c) \cdot \prod_{X_i \in X_R} P(X_i|\mathbf{Y_i}, c), \tag{3}$$

where $\Omega_C$ is the set of possible classes, and $X_R$ the set of observed variables in the network.

## 2.6 Software features

Before initializing RNACOREX, the user must load the engines (miRTarBase, TarBase, DIANA microT-CDS, and TargetScan), as well as the gene annotation database from GENCODE. The package includes a built-in function, `download()`, which must be executed after installation. This function automatically downloads and stores the required resources in a predefined directory. The function `check_engines()` can be used to verify that the databases have been correctly downloaded and stored. If this check returns a positive result, the package is ready to be used.

The MRNC class (MiRNetClassifier) is the main element of RNACOREX. An object of MRNC class is a *scikit-learn* [19] compatible estimator that develops the whole package workflow. This object computes the structural and functional information scores, ranks interactions by their relevance, and builds the CLG model. It also displays the associated coregulation network in an easy and intuitive way.

To initialize an MRNC object, two main objects are required: a $X$ matrix, associated with the mRNA and miRNA expression data, and a $y$ array, corresponding to the phenotype or the class. While data in the $X$ expression matrix should be numerical, the class array should be binary, coded with numerical 0 and 1 values. An MRNC object has five main attributes: `n_con`, `precision`, `mode`, `weight` and `ties`. All five attributes are optional and can be modified by the user. The `n_con` value will correspond to the default number of connections used for fitting the model and displaying the network. The `precision` attribute will define the precision in the conditional mutual information estimation process. `Mode` will be used to select the ranking building approach, being `weight` the weight of the structural information score in the `weighted` case. `Ties` will define the tie-breaking strategy. As a quick start, a network with a specific number of interactions can be fitted with the `fit` method and used for predictions with `predict`. The network can be displayed with

`get_network`. Using `connections_`, the interaction ranking with the associated structural and functional information score of each interaction can be obtained. Further specifications on available methods and default parameters can be found at S3 Text.

## 3 Results

### 3.1 Data

We tested RNACOREX on 13 miRNA and mRNA expression databases obtained from The Cancer Genome Atlas (TCGA) project [20] through the NCI's Genomic Data Commons (GDC) [21] using the University of California Santa Cruz (UCSC) Xena platform [22]. Specifically, the data used was from Breast Invasive Carcinoma (BRCA), Colon Adenocarcinoma (COAD), Head-Neck Squamous Carcinoma (HNSC), Kidney Renal Cell Carcinoma (KIRC), Acute Myeloid Leukemia (LAML), Liver Hepatocellular Carcinoma (LIHC), Lung Adenocarcinoma (LUAD), Lung Squamous Cell Carcinoma (LUSC), Low Grade Glioma (LGG), Sarcoma (SARC), Skin Cutaneous Melanoma (SKCM), Stomach Adenocarcinoma (STAD) and Uterine Corpus Endometrial Carcinoma (UCEC). For model development, only tumoral samples were considered. mRNAs with more than 25% of samples showing fewer than 5 counts, and miRNAs with more than 25% of samples showing fewer than 1 count, were excluded. Differential expression analysis was then performed on the remaining mRNAs retaining only those that were significant after FDR correction ($p.adj < 0.05$). Expression counts were log-normalized as $log_2(x+1)$. Patients were divided into two classes depending on their survival time (long-survival/short-survival) with an equal number of samples in each class. To this end, the 75% of uncensored samples with the shortest overall survival times were assigned to the short-survival class. The remaining 25% of uncensored samples were included in the high-survival class, which was further complemented with censored samples exhibiting the longest follow-up times in order to balance the class sizes. Only censored samples with follow-up times exceeding the maximum survival time observed in the low-survival group were considered for this purpose. RNACOREX was used with two main goals, first, to extract the network most likely associated with the survival time in each disease, and secondly, to evaluate the classification ability of the model.

### 3.2 Model induction

Starting from the most simple network with a single interaction, 200 networks were built for each database, sequentially increasing the complexity of the network by adding a new edge. The network parameters were estimated using a 3-fold CV scheme, implementing three iterations for each network with a different train-test split. The performance of the network with the best metrics computed by RNACOREX was compared to five state-of-the-art classification models. Specifically, the comparison includes two graph-based classification methods, a Graph Kernel approach and a Graph Neural Network, as well as three classical vector-based machine learning models, namely Random Forest, Gradient Boosting, and Support Vector Machines. This selection ensures that both graph-structured and traditional feature-based approaches are represented in the evaluation. At each step, the five classification models were trained using only the variables present in the current network extracted by RNACOREX, ensuring a fair comparison for each specific value of $k$. Fig 2 shows an example of the post-transcriptional network for the COAD database with best classification performance ($k = 97$).

### 3.3 Quantitative performance

Table 1 presents classification performance metrics for the 13 databases analyzed. Values show the best marks out of the 200 runs of the CLG model identified by RNACOREX.

With two balanced classes, the accuracy of the classifier increased from a 9.2% in SKCM to a 24.8% in LAML, compared with random assignment (50% baseline). The behavior of the classifier varied depending on each database, as evidenced by the fact that the network that classified best occurred under different settings; in some cases with many connections (164) and in others with few (45). This may indicate an adaptive behavior that aligns with the biological reality

**Fig 2**. **RNACOREX Post-transcriptional network.** The figure shows the specific network corresponding to the best classification performance obtained for the COAD database, composed of 97 interactions. Green nodes map microRNAs, whereas blue nodes correspond to mRNAs.

underlying the data. Fig 3 presents the distribution of the accuracy for each of the six classification models. When comparing the CLG with the other five models, RNACOREX consistently outperformed other graph-based models (GNN and Graph Kernel), and achieved similar results to vector-based classifiers. Furthermore, RNACOREX offers a significant advantage: its results are fully explainable and the associated coregulation network can be extracted.

In addition to the main results of the article, the performance of RNACOREX was also compared with CGBayesNets [23], a generic Bayesian network package that also implements CLGs. The results showed the advantages of using a pipeline specifically adapted to this problem, over a generic Bayesian network model. All comparisons are included in S6 Text.

### 3.4 TCGA pancancer analysis

To demonstrate the usability of RNACOREX, a preliminary biological analysis of the results was conducted. Based on the data obtained from 13 TCGA databases, the sets of mRNAs, miRNAs, and their interactions comprising the resulting networks were examined. Fig 4 in S5 Text shows the most frequently occurring miRNAs and mRNAs, Fig 4 in S5 Text

**Table 1. Performance.**

| Disease | k | Best Acc. | Best AUC | Mean Acc. (std) | Mean AUC (std) |
|---|---|---|---|---|---|
| BRCA | 124 | 0.683 | 0.693 | 0.639 (0.024) | 0.672 (0.021) |
| COAD | 97 | 0.664 | 0.703 | 0.627 (0.020) | 0.666 (0.023) |
| HNSC | 107 | 0.648 | 0.670 | 0.625 (0.016) | 0.659 (0.015) |
| KIRC | 159 | 0.602 | 0.644 | 0.572 (0.016) | 0.644 (0.024) |
| LAML | 68 | 0.748 | 0.796 | 0.710 (0.025) | 0.771 (0.025) |
| LGG | 45 | 0.685 | 0.735 | 0.667 (0.017) | 0.717 (0.016) |
| LIHC | 62 | 0.677 | 0.711 | 0.646 (0.018) | 0.697 (0.010) |
| LUAD | 56 | 0.669 | 0.704 | 0.642 (0.019) | 0.689 (0.019) |
| LUSC | 73 | 0.634 | 0.685 | 0.611 (0.020) | 0.659 (0.029) |
| SARC | 164 | 0.699 | 0.733 | 0.667 (0.025) | 0.712 (0.021) |
| SKCM | 164 | 0.592 | 0.637 | 0.567 (0.013) | 0.625 (0.008) |
| STAD | 79 | 0.640 | 0.676 | 0.614 (0.016) | 0.662 (0.014) |
| UCEC | 105 | 0.720 | 0.765 | 0.686 (0.025) | 0.741 (0.015) |

**Table notes. k.** Number of interactions of the network with the best accuracy. **Best Acc.** Accuracy of the network. **Best AUC.** AUC of the network with the highest accuracy. **Mean Acc.** Average accuracy (and standard deviation) for all the networks from 1 to 200 interactions. and **Mean AUC.** Average AUC (and standard deviation) for all the networks from 1 to 200 interactions.

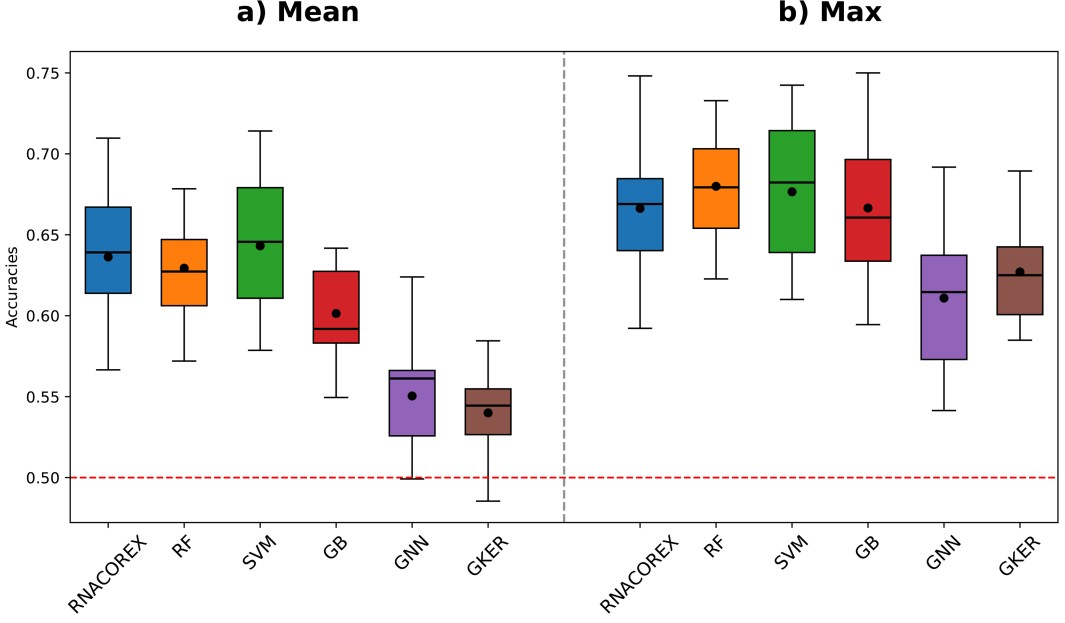

**Fig 3**. **Distribution of accuracies.** The figure shows the distribution of accuracies across all databases for all six classification algorithms: RNACOREX (blue), Gradient Boosting (orange), Random Forest (green), SVM (red), Graph Neural Network (purple) and Graph Kernel (brown). The horizontal red line represents a random assignment (50%) **a) Mean Accuracies.** Shows the distribution of mean accuracies for all *k* values across the 13 databases. **b) Max. Accuracies.** Shows the accuracy distribution of the models including the elements derived from the best network across all databases.

shows the shared interactions across multiple cancers and Fig 4 in S5 Text provides a graphical summary of the number of tissue-specific versus cross-tissue interactions observed in the different networks.

Specifically, hsa-miR-1293 emerged as the most recurrent miRNA in a tissue-specific context, appearing in 54 interactions within the HNSC dataset, being present in approximately half of the interactions in the network. In contrast, hsa-miR-378c was the most widely represented miRNA overall, being involved in a total of 190 interactions across

12 different tissue types. miR-1293 has been previously implicated in head and neck squamous carcinoma [24], while hsa-miR-378c is primarily associated with gastric cancer [25], though it has also been linked to other cancer types [26].

A similar analysis was performed at the mRNA level, revealing BACH2 (9 tissue-specific interactions in COAD) and SLC2A1 and TBL1XR1 (7 interactions in different cancer tissues) as the most recurrent mRNAs. BACH transcription factors are important regulators of pathophysiology in the digestive system and have been associated to colorectal cancers [27]. On the other hand, SLC2A1 has been previously considered as a potential pancancer biomarker [28] and TBL1XR1 has been widely recognised as a protooncogene [29].

At the level of miRNA–mRNA interactions, 3 interactions (hsa-miR-378c - SERPINE1, hsa-miR-378c - SLC2A1 and hsa-miR-4326 - TPX2) appeared as the most frequently observed, appearing all of them in 5 different tissues. While the specific hsa-miR-378c - SERPINE1 interaction has not been associated with cancer, the hsa-miR-378a version of hsa-miR-378c has been directly related with SLC2A1 in cancer contexts [30]. Individually, both SERPINE1 and the previously commented SLC2A1 have been widely associated to cancer [31,32]. TPX2 has also been linked with a variety of cancers [33], but its relation with hsa-miR-4326 remains unknown.

A more detailed discussion of the most relevant findings is provided in *miR-1293 in Head and Neck Squamous Carcinoma* section in S5 Text, emphasizing how these interactions relate to the pathology under study. Overall, this brief analysis highlights the usability of RNACOREX for the biological exploration of complex regulatory networks. The identification of recurrent molecules, as well as previously unreported interactions, underscores the potential of the tool to generate novel biological hypotheses that could guide subsequent experimental validation.

## 4 Availability and future directions

A logical next step in the application of RNACOREX is to focus on specific tissues or conditions, by performing a more in-depth biological analysis of the identified interactions. This will include a systematic functional interpretation, such as pathway and enrichment analyses, to better understand the biological roles of the selected networks in a given context. In addition, incorporating methods to estimate global feature importance including metrics to quantify the overall relevance of individual RNAs or interactions across the network could offer deeper insights into the key biological drivers. Both types of analyses could greatly enhance the relevance of the results and are scheduled for integration into future versions of the package.

One other promising direction is the integration of more detailed coregulatory effects within the interaction networks. Currently, the package assumes a regulatory flow from miRNAs to mRNAs, typically resulting in gene down-regulation due to transcript degradation or translational repression. However, recent evidence suggests that miRNAs can also induce gene up-regulation, particularly through interactions at promoter regions [34,35]. Incorporating this information into RNACOREX would allow users to distinguish between up- and down-regulatory interactions, enhancing the biological interpretability of the resulting networks.

While RNACOREX only allows miRNA-mRNA relations, the incorporation of other forms of regulation, such as mRNA-mRNA (epistatic) or miRNA-miRNA interactions, represents another venue for future development. These additional types of interactions have been linked to various disease processes [36,37], and their inclusion could provide a more comprehensive view of the overall transcriptomic regulatory process.

Extra methods, figures, and tables can be found in S1–S6 texts. The code and the package repository is accessible at https://github.com/digital-medicine-research-group-UNAV/RNACOREX. The package is also available on the Python Package Index (PyPI) at https://pypi.org/project/rnacorex/, allowing direct installation via `pip install rnacorex`. A Docker image of the package including all necessary dependencies for running is also available at https://doi.org/10.5281/zenodo.17397953. The package is open-source with an APACHE 2.0 license.

## Supporting information

**S1 Text. Engines.**
**Table A. Engine nomenclatures.**
**Table B. Engine composition.**
**Fig A. Engine overlap.**
(PDF)

**S2 Text. Theoretical Background.**
**Fig A. CMI Estimations.**
**Fig B. CMI Standard Deviation.**
**Fig C. CMI execution time.**
**Fig D. Tie-breaking procedures.**
(PDF)

**S3 Text. Package Implementation.**
(PDF)

**S4 Text. Experiments.**
**Fig A. Class Assignment.**
**Table A. Dataset characteristics.**
(PDF)

**S5 Text. Results.**
**Table A. Performance.**
**Fig A. Most Repeated RNAs.**
**Fig B. Pancancer Interactions.**
**Fig C. Shared and tissue-specific interactions.**
**Table B. Interaction ranking.**
**Fig D. miR-1293 interactions in HNSC.**
**Fig E. Gene expression levels.**
**Fig F. miR-1293 interacting gene metrics.**
(PDF)

**S6 Text. Benchmarking.**
**Table A. Benchmarking.**
(PDF)

## Author contributions

**Conceptualization:** Aitor Oviedo-Madrid, Ruben Armañanzas.

**Data curation:** Aitor Oviedo-Madrid.

**Formal analysis:** Aitor Oviedo-Madrid.

**Funding acquisition:** Ruben Armañanzas.

**Investigation:** Aitor Oviedo-Madrid, José González-Gomariz.

**Methodology:** Aitor Oviedo-Madrid, José González-Gomariz, Ruben Armañanzas.

**Project administration:** Aitor Oviedo-Madrid.

**Resources:** Ruben Armañanzas.

**Software:** Aitor Oviedo-Madrid.

**Supervision:** José González-Gomariz, Ruben Armañanzas.

**Validation:** Aitor Oviedo-Madrid.

**Visualization:** Aitor Oviedo-Madrid, José González-Gomariz.

**Writing – original draft:** Aitor Oviedo-Madrid, José González-Gomariz.

**Writing – review & editing:** Aitor Oviedo-Madrid, Ruben Armañanzas.

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
