## [Decision Letter · Decision Letter 0]

13 May 2025

PCOMPBIOL-D-24-01999

RNACOREX - RNA coregulatory network explorer and classifier

PLOS Computational Biology

Dear Dr. Oviedo-Madrid,

Thank you for submitting your manuscript to PLOS Computational Biology. After careful consideration, we feel that it has merit but does not fully meet PLOS Computational Biology's publication criteria as it currently stands. Therefore, we invite you to submit a revised version of the manuscript that addresses the points raised during the review process.

Please submit your revised manuscript within 60 days Jul 13 2025 11:59PM. If you will need more time than this to complete your revisions, please reply to this message or contact the journal office at ploscompbiol@plos.org. Please include the following items when submitting your revised manuscript:

We look forward to receiving your revised manuscript.

Kind regards,

Mark Alber, Ph.D.

Section Editor

PLOS Computational Biology

**Journal Requirements:**

4) We have noticed that you have uploaded Supporting Information files, but you have not included a complete list of legends. Please add a full list of legends for your Supporting Information files ( RNACOREX-main.zip) after the references list.

Potential Copyright Issues:

i) Figure 1. Please confirm whether you drew the images / clip-art within the figure panels by hand. If you did not draw the images, please provide (a) a link to the source of the images or icons and their license / terms of use; or (b) written permission from the copyright holder to publish the images or icons under our CC BY 4.0 license. Alternatively, you may replace the images with open source alternatives. See these open source resources you may use to replace images / clip-art:

ii) The following file contains screenshots: Library_instructions.pdf in RNACOREX-main.zip. We are not permitted to publish these under our CC-BY 4.0 license, websites are usually intellectual property and are copyrighted.This includes peripheral graphics of the web browser such as icons and button. We ask that you please remove or replace it.

**Reviewers' comments:**

Reviewer's Responses to Questions

Reviewer #1: The authors present RNACOREX, a computational method for analyzing miRNA-mRNA interactions using a combination of structural information from existing databases and functional information derived from expression data. The methodology culminates in the construction of a Conditional Linear Gaussian (CLG) classifier.

The comments will be sorted by the supplementary material sections as look like an extended version of the main manuscript.

Section 2.1

The structural information score aims to minimize false interactions resulting from the Open World Assumption. One of its key components, the consistency index, evaluates whether a given interaction is reported in one, two, or all three databases. However, one of these databases exclusively contains experimentally validated miRNA–target interactions. The presence of an interaction in this database should be interpreted as biologically confirmed rather than merely inferred.

How is this aspect incorporated into the scoring mechanism? Additionally, how many interactions present in miRTarBase have been discarded or assigned a low score by this method?

The equation (2) for the structural information score lacks clarity in how exactly the normalized consistency index contributes to the final score. The relationship between I_norm(A,B) and w_ij^ab should be explicitly defined.

Also, RNACOREX seems to consider only the already existent pairs from other databases, excluding the discovery of newly interactions.The authors should clarify whether the filtering process might introduce biases by potentially excluding novel but biologically relevant interactions not yet documented in databases.

Section 2.2

While kernel density estimation is a reasonable approach, the authors should address how they handle the curse of dimensionality, particularly for datasets with limited samples.

The default of 10 trapezoids for numerical integration seems arbitrary. The authors should provide justification for this choice or demonstrate its adequacy through validation experiments.

Section 2.3

The alternating selection approach from structural and functional rankings is interesting but lacks theoretical justification. The authors should explain why this approach is preferable to other ranking combination methods (e.g., weighted combinations).

The method for handling ties in either ranking is not specified but could significantly impact results.

Section 2.4

Figure 2 illustrates the post-transcriptional network generated. Two main types of interactions can be identified:

1.Those with higher Structural Information than Functional Information

2.Those with the inverse pattern

Is there a systematic bias favoring structural or functional information during training? If so, how does this affect prediction accuracy? It might also be useful to check feature importance scores from the classifier to support this claim.

Section 3

A Docker container with all necessary data and a well-organized folder structure would significantly enhance usability, enabling users to seamlessly run the methods and better understand the required inputs.

Section 4.2

The differential expression analysis (DEA) between Short and Long Survival groups is used to reduce the number of mRNA candidates before constructing the network. This is a reasonable approach to remove non-variable mRNAs that are unlikely to influence overall survival.

However, filtering by p-value instead of FDR is statistically incorrect when applying thousands of tests. Moreover, the justification for further filtering—"genes are filtered by RNACOREX using databases of predicted and validated interactions"—raises concerns about the DEA approach. If RNACOREX is already filtering mRNAs based on documented interactions, why apply a questionable pre-filter beforehand?

Additionally:

Supplementary Figure 3 is not properly referenced.

Supplementary Figure 3 contains a typo ("enaugh" → "enough").

Section 4.3

While Random Forest (RF), Gradient Boosting (GB), and Support Vector Machine (SVM) are widely used in vector-based classification, they are not inherently designed for graph-based classification.

To establish a baseline for evaluating RNACOREX, I suggest comparing it against more graph-specific state-of-the-art methods, such as:

Graph Neural Networks (GNNs)

Graph Kernels (e.g., Weisfeiler-Lehman kernel + SVM)

Section 5.1

The section begins by claiming a 10%–30% accuracy increase, but this should be clearly visualized in a single figure.

Additionally, does accuracy correlate with the number of mRNAs used? At first glance, datasets with the highest performance (LGG and UCEC, with 4,729 and 3,000 mRNAs, respectively) contain substantially more mRNAs than the lowest-performing datasets (SKMC and LUSC, with 1,270 and 1,079 mRNAs). Is there a statistical correlation (spearman) between accuracy and mRNA count?

Other figure-related concerns:

1.Figures for different sample results (Supp. Fig. 4–15) lack proper descriptions.

What is the y-axis?

What does the blue ribbon represent?

What do the rank and metrics indicate?

2.Since "Test" remains constant across three figures, and the different models are color-coded, it would be more readable if all lines were presented within a single graph.

Section 5.2

The final validation of the tool is underdeveloped. Given that miRNA–mRNA interactions are cancer-specific and focus on overall survival shifts, I would expect an analysis demonstrating that the mRNAs in predicted miRNA–mRNA pairs are linked to biologically relevant pathways.

I strongly suggest performing such an enrichment/pathway analysis to better showcase the tool’s potential.

Additionally:

Supplementary Figures 17–29 add little to no value, as they are neither analyzed nor referenced.

miRTarBase was updated on January 6, 2025—newly added interactions could be used to validate the predictions.

Overall Questions:

1.Why not leverage TCGA’s paired tumor–healthy data?

TCGA provides a well-characterized, paired dataset, facilitating a clearer comparison between tumor and normal samples.

Why did the authors opt for a tumor-specific survival drift approach rather than identifying miRNA–mRNA interactions that directly drive the oncological phenotype?

Would integrating paired tumor-normal data improve the model’s interpretability?

2.Explanatory Capability

The abstract highlights RNACOREX's "inherent explanatory capability related to the pathology being studied", yet no supporting results are provided.

No bibliographical validation is offered to demonstrate that the predicted miRNA–mRNA pairs correlate with changes in overall survival across different cancers.

Reviewer #2: RNACOREX was developed for the classification capability for cancer survival based on miRNA-mRNA interactions and their functional relevance within a Conditional Linear Gaussian (CLG) Bayesian network framework. However, some aspects of the methodology and presentation require clarification and more detailed justification before publication. Key details regarding the implementation of the structural and functional information scoring, the network learning process within the CLG framework, and a more in-depth discussion of the biological interpretation of the results and network structures are needed. The performance evaluation could also benefit from additional context and possibly alternative metrics.

Major revision

1. The manuscript would benefit significantly from a more detailed, practical case study. Currently, the authors highlight a few pan-cancer miRNA biomarkers. However, deeper biological insight into specific miRNA-mRNA interactions is lacking. I suggest the inclusion of a case study focused on a single cancer type, analyzing how the top-ranked interactions relate to known biological pathways or potentially novel mechanisms. A deeper biological analysis of the network structures and the roles of the highlighted interactions would significantly strengthen the manuscript's impact.

2. The manuscript would benefit from specifying the advantages of MTI interaction in survival prediction over simply a group of miRNAs or a group of mRNAs.

3. The performance or novelty of RNACOREX should be compared with previous methods such as CGBayesNets; miRNA–Target Gene Regulatory Networks: A Bayesian Integrative Approach to Biomarker Selection with Application to Kidney Cancer; A Bayesian framework that integrates multi-omics data and gene networks predicts risk genes from schizophrenia GWAS data and so on.

4. The use of the Conditional Linear Gaussian (CLG) classifier is briefly justified by stating that it is suitable for gene expression data. A clearer theoretical rationale for its choice—possibly compared to alternative probabilistic models—would improve the manuscript's methodological transparency. Please clarify the process of learning the CLG Bayesian network structure. Is a structure learning algorithm employed, and if so, which one and how does the interaction ranking influence it? How are the parameters of the CLG learned from the data?

5. Related to 4, key methodological details—such as data preprocessing steps (Supplementary Material Section 4.2 is referenced for preprocessing, but a brief mention in the main text would improve clarity), methods for dividing patients based on survival time (long/short) (how this cutoff was determined). A brief overview of these steps should be included in the main text for clarity and transparency.

6. While the manuscript describes the general idea of structural and functional information scores and an interaction hierarchy, the specific mathematical formulations and weighting schemes are referenced to the Supplementary Material. These are fundamental to the RNACOREX methodology and should be clearly summarized in the main manuscript or provided with sufficient detail in the supplementary material (once accessible) to allow for critical evaluation and reproduction.

7. "The performance of the network with the best metrics computed by RNACOREX was compared to three state-of-the-art classification models: Random Forest, Gradient Boosting, and SVM." Please provide proper reference or rationale that these are considered state-of-the-art for this problem type. In addition, Table 1 only shows RNACOREX's performance for a subset of the datasets. A comprehensive comparison, presenting the key performance metrics for the mentioned models across the selected datasets, is necessary to allow readers to assess the relative performance of RNACOREX. A comprehensive figure would be much stronger to support your claims. It would also strengthen the manuscript to include the full performance comparison in the main text, rather than relegating it to supplementary materials.

8. While the supplementary files are linked at the end of the manuscript, they are either missing from the GitHub repository or difficult to locate there. The wording in the manuscript gives the impression that the files are available on GitHub, which may cause readers to overlook the fact that they are actually included at the end of the manuscript.

Miner revision

9. Figure 1 provides a workflow, which is helpful. Improving the structure and clarifying the meaning of figures would benefit the manuscript.

10. While the manuscript is generally well written, there are several imprecise sentences. I recommend a thorough proofreading and language polishing to improve fluency and clarity.

11. Terms like “post-transcriptional network,” “coregulatory network,” and “regulatory network” are used interchangeably. Consider being consistent in definitions throughout.

12. For readers less familiar with classification metrics, consider briefly explaining key metrics such as AUC, sensitivity, and specificity in the manuscript.

13. To promote adoption of RNACOREX, consider uploading the package to the PyPI repository, allowing users to install it via pip. This would eliminate the need for manual environment configuration or file downloads.

Reviewer #3: Authors propose a Python package that finds disease associated post-transcriptional networks and use them to classify micro and mRNA interactions.

I have the following concerns and comments.

My main concern is that the authors propose a new method based on some heuristics but the method is not benchmarked against any existing methods. THe paper is positioned as a software paper, a Pyhthon package certain features, but the underlying method's validity is not well established. It is not clear if as a user we are using the state of the art method with nice software features.

Authors claim the following but there is no experimental support on this: "However, not all miRNAs interact with all mRNAs, and relying solely on expression data often results in a high number of false positives. RNACOREX uses a hybrid approach... This strategy improves accuracy and reduces the likelihood of misleading results..." Please provide evidence that the first approach has many false positives and this new strategy improves accuracy.

The abstract is confusing with a poor flow. It needs to be restructured.

For instance, not clear at all what this means: "The process is developed using a hybrid approach by combining expert information with empirical knowledge extracted from expression data"

"RNACOREX is a new, easy-to-use tool that allows researchers to find disease associated post-transcriptional networks and use them to classify new unseen observations of micro and mRNA quantifications." Not clear what is classified yet. This is later clarified in classifying tumors. Also micro -> micro RNA.

Section 2.1 aims to provide an overview but it rather is confusing. Figure 1 and its caption is also confusing. It is not clear what structural information refers to and what is expert information-based score. Section 2.2 explains structural information score but I do not understand what this needs to do with structure it seems to be a support based on number of databases the interaction occurs and the confidences of those supports.

Authors divide patients into two classes depending on their survival time (long-survival vs. short-survival) but ensure that there are equal number of samples in each class. Depending on the underlying distribution this may make the time periods meaningless if there is an imbalance in the data. Is this the case?

Not clear what this means: "The first two databases include only predicted interactions based on their genomics".

L82 first ->First

**Have the authors made all data and (if applicable) computational code underlying the findings in their manuscript fully available?**

Reviewer #1: Yes

Reviewer #2: Yes

Reviewer #3: Yes

PLOS authors have the option to publish the peer review history of their article (what does this mean?). If published, this will include your full peer review and any attached files.

Reviewer #1: No

Reviewer #2: No

Reviewer #3: No

**Figure resubmission:**
---

## [Decision Letter · Decision Letter 1]

24 Aug 2025

PCOMPBIOL-D-24-01999R1

RNACOREX - RNA Coregulatory Network Explorer and Classifier

PLOS Computational Biology

Dear Dr. Oviedo-Madrid,

Thank you for submitting your manuscript to PLOS Computational Biology. After careful consideration, we feel that it has merit but does not fully meet PLOS Computational Biology's publication criteria as it currently stands. Therefore, we invite you to submit a revised version of the manuscript that addresses the points raised during the review process.

Please submit your revised manuscript within 60 days Oct 24 2025 11:59PM. If you will need more time than this to complete your revisions, please reply to this message or contact the journal office at ploscompbiol@plos.org. Please include the following items when submitting your revised manuscript:

We look forward to receiving your revised manuscript.

Kind regards,

Mark Alber, Ph.D.

Section Editor

PLOS Computational Biology

Mark Alber

Section Editor

PLOS Computational Biology

**Journal Requirements:**

**Reviewers' comments:**

Reviewer's Responses to Questions

**Comments to the Authors:**

Reviewer #1: I want to thank the authors for taking the time to satisfactorily address the comments I raised. After reading the software submission requirements, I understand that an exhaustive biological analysis is not necessary. Even so, I am glad that the other reviewers also requested it. We are not asking for a complex experiment to prove that the predicted interactions are physically occurring, but rather for some interpretability in the model — for example, assessing whether the predicted interactions are related to the phenotypes under study. In the same way the DEG pipeline is proposed, it would be useful to verify that the model is capturing relevant biological information rather than purely topological patterns.

Reviewer #3: I thank the authors for addressing my comments. However, some of the issues I raised still remain.

-Regarding my question on benchmarking the pipeline against others (Q1), the authors reckon that often a method does either network construction or classification. While it is still possible for authors to compare with such methods, they combine multiple tasks and it is understandable to just discuss this issue in the paper.

-Authors indicate that there is only one method which combines network inference and classification: CGBayesNets. It is different than RNACOREX because it does not include curated. Authors indicate that they are not comparable. I do not understand why this is a problem. CGBayesNets aims to solve the same problem with less data and even uses CLG. So this comparison should show the importance of the prior information and the contribution of the authors to justify the need for a new pipeline.

Minor.

- Regarding my comment (Q2), could the authors cite references from the literature to back up the claim that "It is well established in the literature that the identification of false positives is a recurrent and significant challenge when working with miRNA–mRNA interaction predictions".

- The issue in Section 2.1 remains: It is not clear what "structural information" is at this point in "Candidate interactions are initially filtered based on their structural information". This sounds like the 2D/3D structure of the RNAs but it is not and it is confusing. Please clarify in text with you answer to Q5.

- Abstract: while offers —> while offering

Reviewer #4: The paper introduces RNACOREX, a graph-based framework that integrates known miRNA–mRNA interaction databases with expression data to build coregulation networks. Using a Conditional Linear Gaussian classifier, RNACOREX classifies patient samples while simultaneously highlighting potentially disease-relevant regulatory interactions. The method allows users to control network construction through different interaction-ranking strategies and number of edges, balancing predictive accuracy with biological interpretability.

Major Issue 1: The performance evaluation of RNACOREX may be confounded by the fact that Differential Expression Analysis (DEA) is performed using the long and short survival group labels. Since DEA already incorporates class information, the subsequent network construction cannot fully demonstrate that the model itself contributes to classification performance. In principle, one could randomly select mRNAs, perform DEA, and then use a standard classifier (e.g., SVM) to obtain comparably good results. The authors should clarify how the results would change if DEA were not performed as a preprocessing step, or provide an analysis demonstrating that the observed performance is not solely driven by the initial DEA filtering.

Major Issue 2: In the introduction, the authors do not clearly articulate the primary objective of RNACOREX. Classifying patients according to survival time should be presented as a validation strategy rather than as the central aim of the model. Based on the structure of the paper, it appears that the ultimate goal is to identify miRNA–mRNA interactions and potential gene groups associated with specific diseases. If this interpretation is correct, the authors should explicitly state this goal in the abstract/introduction to better frame the contribution and novelty of the work.

Major Issue 3: The claim of “successfully uncovering disease-specific networks” appears overstated. As reported, the model’s performance is only comparable to, or in some cases weaker than, existing machine learning approaches. Moreover, the current implementation of RNACOREX does not yet provide substantive insights into underlying pathology. The authors should temper this statement and clarify that such interpretability remains a potential future direction rather than a current strength of the model.

Minor Issue 1: RNACOREX integrates four datasets, but the manuscript does not explain how potential bias or batch effects across these datasets are addressed. Furthermore, it would be informative if the authors could report how many miRNA–mRNA interactions are shared across datasets and how many are unique to each source. This additional analysis would provide users with a clearer understanding of the consistency and complementarity of the datasets employed.

Minor Issue 2: In line 142, the authors state that “only those interactions that are supported by at least one of the previously mentioned sources” are retained. However, in the definition of the SI score (line 186), it is stated that “the interaction must be present in at least two databases to be considered.” Does this mean that SI scores are only computed for interactions present in at least two sources, while interactions supported by only one source are assigned only a functional information score without any SI score? Clarification is needed.

Minor Issue 3: In line 217, the authors describe a tie-breaking strategy. Have alternative tie-breaking methods been considered or tested, and if so, how do the results compare?

Minor Issue 4: In the Conditional Linear Gaussian Classifier section (line 250), the class variable C is not clearly defined. This should be explicitly described in the main text rather than left only to the supplementary material.

Minor Issue 5: Line 352 mentions that the model was trained using a 3-fold cross-validation. Have the authors considered alternative cross-validation strategies, and what was the rationale for choosing 3-fold in particular?

**Have the authors made all data and (if applicable) computational code underlying the findings in their manuscript fully available?**

Reviewer #1: Yes

Reviewer #3: Yes

Reviewer #4: Yes

PLOS authors have the option to publish the peer review history of their article (what does this mean?). If published, this will include your full peer review and any attached files.

Reviewer #1: No

Reviewer #3: No

Reviewer #4: **Yes: **Zhana Duren

**Figure resubmission:**
---

## [Decision Letter · Decision Letter 2]

24 Oct 2025

Dear Oviedo-Madrid,

We are pleased to inform you that your manuscript 'RNACOREX - RNA Coregulatory Network Explorer and Classifier' has been provisionally accepted for publication in PLOS Computational Biology.

Best regards,

Mark Alber, Ph.D.

Section Editor

PLOS Computational Biology

Mark Alber

Section Editor

PLOS Computational Biology

Reviewer's Responses to Questions

**Comments to the Authors:**

Reviewer #4: Thank authors. My comments are addressed.

**Have the authors made all data and (if applicable) computational code underlying the findings in their manuscript fully available?**

Reviewer #4: Yes

PLOS authors have the option to publish the peer review history of their article (what does this mean?). If published, this will include your full peer review and any attached files.

Reviewer #4: No

---

## [Editor Report · Acceptance letter]

PCOMPBIOL-D-24-01999R2

RNACOREX - RNA Coregulatory Network Explorer and Classifier

Dear Dr Oviedo-Madrid,

I am pleased to inform you that your manuscript has been formally accepted for publication in PLOS Computational Biology. Your manuscript is now with our production department and you will be notified of the publication date in due course.

With kind regards,

Anita Estes
